# Plasma-Induced Oxidation Products of (–)-Epigallocatechin Gallate with Digestive Enzymes Inhibitory Effects

**DOI:** 10.3390/molecules26195799

**Published:** 2021-09-24

**Authors:** Gyeong Han Jeong, Tae Hoon Kim

**Affiliations:** Department of Food Science and Biotechnology, Daegu University, Gyeongsan 38453, Korea; jkh4598@hanmail.net

**Keywords:** (−)-epigallocatechin gallate, non-thermal processing, dielectric barrier discharge plasma, α-glucosidase, α-amylase

## Abstract

(−)-Epigallocatechin gallate (EGCG), the chief dietary constituent in green tea (*Camellia sinensis*), is relatively unstable under oxidative conditions. This study evaluated the use of non-thermal dielectric barrier discharge (DBD) plasma to improve the anti-digestive enzyme capacities of EGCG oxidation products. Pure EGCG was dissolved in an aqueous solution and irradiated with DBD plasma for 20, 40, and 60 min. The reactant, irradiated for 60 min, exhibited improved inhibitory properties against α-glucosidase and α-amylase compared with the parent EGCG. The chemical structures of these oxidation products **1**–**3** from the EGCG, irradiated with the plasma for 60 min, were characterized using spectroscopic methods. Among the oxidation products, EGCG quinone dimer A (**1**) showed the most potent inhibitory effects toward α-glucosidase and α-amylase with IC_50_ values of 15.9 ± 0.3 and 18.7 ± 0.3 μM, respectively. These values were significantly higher than that of the positive control, acarbose. Compound **1**, which was the most active, was the most abundant in the plasma-irradiated reactant for 60 min according to quantitative high-performance liquid chromatography analysis. These results suggest that the increased biological capacity of EGCG can be attributed to the structural changes to EGCG in H_2_O, induced by cold plasma irradiation.

## 1. Introduction

Diabetes mellitus (DM) is one of the most common chronic diseases worldwide. It is a polygenic complex metabolic disorder caused by high blood glucose levels. DM is characterized by serious complications, such as diabetic neuropathy, stroke, peripheral vascular disease, renal failure, and blindness, resulting in reduced life expectancy, increased disability, and enormous health costs [1]. The most effective therapy for type 2 diabetes mellitus (T2DM) is to control optimal blood glucose levels after meals [2]. α-glucosidase and α-amylase secreted from the small intestine and pancreas, respectively, are digestive enzymes that break down monosaccharides. These enzymes delay glucose absorption and lower postprandial blood glucose levels, making them valuable therapeutic methods for the treatment of T2DM in humans. Despite numerous studies on the development of α-glucosidase and α-amylase inhibitors, such as acarbose and voglibose, only a few are currently available. Most sugar mimics require tedious preparation steps, which can lead to serious gastrointestinal side effects [3]. Therefore, the development of new bioactive ingredients with potential α-glucosidase and α-amylase inhibitory ingredients from natural sources has attracted considerable attention in the food industry [4].

Plasma is an ionized gas generated by electric discharge and exerts non-thermal sterilization process [5]. Among the recently developed types of non-thermal plasma processing apparatuses, dielectric barrier discharge (DBD) plasma generates plasma at low temperatures by discharging a voltage between dielectric barriers under atmospheric pressure [6]. DBD plasma is one of the most popular tools to generate and affordable by power supply [7]. The generated plasma produces large amounts of ions, electric discharge, ozone, UV photons, and various reactive oxygen species—which can inactivate microorganisms—and has been demonstrated to be an advanced non-thermal technology applied in food processing [8,9]. A recent study reported the development of newly generated products induced by the DBD plasma irradiation of natural polyphenolic compounds. In addition, a DBD plasma treatment under alcoholic solution conditions was readily converted to methylene-linked dimeric products (*trans*-resveratrol, sesamol, and phloridzin), which showed improved pharmacological activity compared with the parent compounds [10,11,12,13]. On the other hand, its influences on plasma-induced oxidation under aqueous conditions, associated with changes in the chemical structure and biological activities, are not completely understood.

(−)-Epigallocatechin gallate (EGCG) is a major bioactive component isolated from the leaves of *Camellia sinensis* var. *sinensis* [14] that has been reported to have a range of pharmacological effects, such as anti-aging, anti-diabetic, anti-bacterial, hypocholesterolemic, antioxidant, and radical scavenging properties [15]. EGCG, however, is quite unstable under oxidative conditions. Several studies have examined the effects of structural transformations and enhancements on the biological properties of EGCG using various physicochemical methods [16,17,18]. The present study examined the influences of non-thermal DBD plasma irradiation on the chemical structures and biological activity of EGCG under H_2_O conditions. The newly formed products showed improved hypoglycaemic effects against α-glucosidase and α-amylase compared with the parent EGCG. This paper tentatively proposes a plausible mechanism for the oxidation pathway of newly generated products **1**–**3** produced after plasma irradiation of EGCG.

## 2. Results and Discussion

### 2.1. Characterization of Oxidation Products

Pure EGCG dissolved in aqueous solution was exposed directly for 20, 40, and 60 min, and the structures of oxidation products were determined using spectroscopic methods (Figure 1). HPLC. The reaction mixture displayed three newly generated peaks in the HPLC chromatogram as transformed products from EGCG (Figure 2). Among the dried reactant, a sample solution containing EGCG that was irradiated for 60 min showed the most enhanced anti-diabetic effects on α-glucosidase and α-amylase inhibitions compared with that of the parent EGCG (Table 1). Successive column chromatography isolation of the 60 min-treated EGCG reactant led to the isolation of three oxidized products, **1**–**3**. The purified compounds were identified as EGCG quinone dimer A (**1**), theacitrinin A (**2**), and gallic acid (**3**), through a comparison of their spectroscopic data (NMR and FABMS) with the literature values. The chemical structural characteristics are detailed below. Compound **1**: Brown amorphous powder, [α]^25^_D_ −62.3 [*c* 0.1, MeOH (1 mg/mL)], FABMS *m*/*z* 929 [M-H]^−^, ^1^H NMR (500 MHz, CD_3_OD): *δ* 6.92 (4H, s, H-2″, 6″), 6.54 (2H, d, *J* = 1.0 Hz, H-6′), 6.03 (2H, d, *J* = 2.0 Hz, H-8), 5.99 (2H, d, *J* = 2.0 Hz, H-6), 5.68 (2H, m, H-3), 4.56 (2H, br s, H-2), 3.22 (2H, br s, H-2′), 2.95 (2H, m, H-4), 2.93 (2H, m, H-4), ^13^C NMR (125 MHz, CD_3_OD): *δ* 196.5 (C-5′), 165.9 (C-7″), 156.6 (C-7), 156.5 (C-9), 155.8 (C-1′), 154.4 (C-5), 144.9 (C-3″, 5″), 138.5 (C-4″), 126.7 (C-6′), 119.6 (C-1″), 108.9 (C-2″, 6″), 103.7 (C-3′), 97.7 (C-10), 95.6 (C-8), 94.4 (C-6), 84.7 (C-4′), 75.9 (C-2), 63.8 (C-3), 59.2 (C-2′), 25.2 (C-4). All spectral data were consistent with those describing EGCG quinone dimer A [19], and compound **1** was identified as EGCG quinone dimer A (Figure 1).

Compound **2**: Brown amorphous powder, [α]^25^_D_ +104.8 [*c* 0.1, MeOH (1 mg/mL)], FABMS *m*/*z* 579 [M-H]^−^, ^1^H NMR (500 MHz, CD_3_OD): *δ* 6.95 (2H, s, H-2′, 6′), 6.71 (1H, s, H-f), 6.05 (1H, d, *J* = 2.0 Hz, H-8), 5.98 (1H, d, *J* = 2.0 Hz, H-6), 5.85 (1H, br s, H-c), 5.73 (1H, m, H-3), 5.23 (1H, br s, H-2), 4.35 (1H, s, H-e), 2.91 (1H, dd, *J* = 17.0, 4.0 Hz, H-4), 2.80 (1H, dd, *J* = 17.0, 1.0 Hz, H-4), ^13^C NMR (125 MHz, CD_3_OD): *δ* 200.1 (C-b), 193.9 (C-l), 174.0 (C-d), 166.0 (C-7′), 157.0 (C-g), 156.9 (C-5), 156.1 (C-7), 155.0 (C-9), 148.2 (C-3′, 5′), 145.1 (C-i), 143.0 (C-k), 138.9 (C-4′), 134.0 (C-h), 125.0 (C-c), 122.0 (C-1′), 115.0 (C-j), 108.8 (C-2′, 6′), 106.1 (C-f), 100.4 (C-10), 97.2 (C-6), 95.9 (C-8), 84.8 (C-a), 73.1 (C-2), 63.0 (C-3), 54.8 (C-e), 29.0 (C-4). All spectral data were consistent with those describing theacitrinin A [20], and compound **2** was identified as theacitrinin A (Figure 1).

Compound **3**: White amorphous powder, ^1^H NMR (500 MHz, CD_3_OD): *δ* 7.08 (2H, s, H-2, 6). All spectral data were consistent with those describing gallic acid [21], and compound **3** was identified as gallic acid (Figure 1).

### 2.2. Inhibitory Effects of α-Glucosidase and α-Amylase

Managing the glucose level in postprandial plasma is effective in the early treatment of diabetes. Inhibition of digestive enzymes, such as α-glucosidase and α-amylase, which play a role in carbohydrate digestion, is an important strategy for decreasing postprandial hyperglycemia [22]. According to several in vivo studies, α-glucosidase and α-amylase inhibition are among the most effective diabetes treatments [23]. The enzyme-inhibitory activities of plasma-treated EGCG in a time-dependent manner increased after exposure to 60 min of non-thermal plasma compared with pure EGCG (Table 1). All isolated oxidation products **1**–**3**, produced by irradiating EGCG for 60 min, were assessed for their hypoglycaemic effects against α-glucosidase and α-amylase inhibition assays, using acarbose as the positive control.

The plasma-irradiated EGCG in an H_2_O solution for 60 min exhibited the highest α-glucosidase inhibitory effects, with an IC_50_ value of 20.1 ± 0.6 μg/mL, lower than the standard EGCG (IC_50_: 44.5 ± 1.2 μg/mL) (Table 1). The α-glucosidase inhibitory activity of oxidized EGCG dimer **1**, which possessed a 3,4-dihydroxylated 3-oxygen-bridged six-membered ring moiety at the B-ring, was most potent in this bioassay system with an IC_50_ value of 15.9 ± 0.3 μM, which was lower than the positive control acarbose (IC_50_: 169.0 ± 1.8 μM). Theacitrinin A (**2**), in which 2,5-cyclohexadien-1-one at the EGCG B-ring was aromatized, was relatively less potent, with an IC_50_ value of 20.7 ± 0.5 μM. The isolated simple phenolic acid (**3**) showed no discernible activity, with an IC_50_ value of >300 μM (Table 2).

In addition, the α-amylase inhibitory capacities of the 60 min-irradiated reactants were improved significantly with an IC_50_ value of 16.5 ± 0.7 μg/mL compared with that of EGCG (IC_50_: 34.5 ± 0.9 μg/mL) (Table 1). Among the isolated compounds, oxidized flavan 3-ol derivatives **1** and **2** showed significant potent inhibitory effects in the α-amylase inhibition assay, with IC_50_ values of 18.7 ± 0.3 μM and 25.3 ± 0.6 μM, respectively, compared with the positive control acarbose (IC_50_: 70.7 ± 1.4 μM) (Table 2). EGCG quinone dimer A was first formed and identified from EGCG oxidized by a peroxyl radical reaction [19,24]. In addition, EGCG quinone dimer A and theacitrinin A were produced from EGCG during an enzymatic oxidation process by polyphenol oxidase (Figure 3) [20,25,26,27]. Interestingly, there has been no research on the biological efficacy of EGCG quinone dimer A and theacitrinin A. To the best of the authors’ knowledge, this is the first example of oxidation in an H_2_O solution of EGCG quinone dimer A (**1**), theacitrinin A (**2**), and gallic acid (**3**) by non-thermal plasma processing, which has beneficial anti-diabetic activities.

### 2.3. Quantitative Analysis of Newly Generated Products

The absolute contents of the isolated compounds in the plasma-irradiated EGCG of 20, 40, and 60 min were quantified using an external standard method, as shown in Figure 2 and Table 3. To further evaluate the relationship between the anti-diabetic effects and the composition of the oxidized mixture, the active compound contents were identified by HPLC using a UV-PDA detector. Contents of the isolated compounds in the 20, 40, and 60 min-irradiated EGCG were quantified using the external standard method; the results are shown in Table 3. Five concentration points (*n* = 5) were used for the preparation of the calibrations curve, and the calibration curve of the pure solution of the standard compounds was completely linear (*R*^2^ > 0.999) (Appendix A). The retention times (*t*_R_) of the newly formed EGCG quinone dimer A (**1**) (*t*_R_ 15.0 min), theacitrinin A (**2**) (*t*_R_ 16.2 min), and gallic acid (**3**) (*t*_R_ 8.0 min) were detected by plasma-irradiated EGCG (*t*_R_ 15.6 min) for three different times. Quantitative analysis showed that the contents of the most potent EGCG quinone dimer A (**1**) in the plasma-irradiated EGCG at 20, 40, and 60 min were 126.0 ± 0.9, 266.7 ± 1.1, and 321.8 ± 1.2 mg/g, respectively, which is in accordance with the enhanced hypoglycaemic effects of each oxidized mixture (Table 3).

The ionized quasi-neutral gas of DBD, plasma-induced in ambient air, was reported to be comprised of UV photons, ions, atomic species, and reactive oxygen species (ROS) [8]. In particular, generated ROS and free radicals can inactivate microorganisms, and have attracted interest in the advanced non-thermal sterilization processing of foods [28]. In addition, free radicals are powerful oxidizing agents, whereas aqueous electrons and hydrogen atoms are reducing agents. These results suggest that reactive molecular species and free radicals might be induced in aqueous solutions and are capable of dimerization and degradation of the chemical structure of EGCG, resulting in generated oxidation products, such as EGCG quinone dimer A (**1**), theacitrinin A (**2**), and gallic acid (**3**). In this mechanism, EGCG quinone dimer A (**1**) is produced as a major product by stereoselective dimerization of the EGCG quinone (Figure 3). These results can be applied to the food industry to improve safety and quality, and can enhance biological activity through the application of plasma-processing technology.

## 3. Materials and Methods

### 3.1. General Experimental Procedures

(−)-Epigallocatechin gallate (EGCG), acarbose, starch, acetonitrile (MeCN), α-glucosidase (EC 3.2.1.20) from *Saccharomyces cerevisiae*, α-amylase (EC 3.2.1.1) from a porcine pancreas, and *p*-nitrophenyl-α-_D_-glucopyranoside (*p*-NPG) were obtained from Sigma-Aldrich (St. Louis, MO, USA). ^1^H-, ^13^C-nuclear magnetic resonance (NMR), heteronuclear single quantum correlation (HSQC), and heteronuclear multiple bond correlation (HMBC) spectroscopy were measured on a Bruker Avance NEO-500 spectrometer (500 MHz, Bruker, Karlsruhe, Germany), with CD_3_OD (*δ*_H_ 3.35, *δ*_C_ 49.0) as a solvent and tetramethylsilane (TMS) as an internal standard. The specific optical rotation was obtained using a JASCO P-2000 polarimeter (JASCO, Tokyo, Japan), and fast atom bombardment mass spectrometry (FABMS) was recorded on a JMS-700 spectrometer (JEOL, Tokyo, Japan) in negative mode.

### 3.2. Nonthermal Plasma Irradiation

The dielectric barrier discharge (DBD) treatment device used was installed on the inner walls of a parallelepiped plastic chamber (150 mm × 150 mm × 275 mm) [29]. The DBD apparatus was composed of four surface DBD sources. Each source was comprised of a 0.6 mm-thick fused silica plate, 100 mm × 100 mm in size. The power supply consisted of an arbitrary waveform generator (Tektronix AFG3021 C). A sinusoidal waveform with a frequency 2.5 kHz and an amplitude voltage of 4 kV was applied between the two electrodes during the operation. The dissipated power by plasma was 65 (± 5%) W, and the temperature increased from 20 °C to 40 °C during the operation [29]. The EGCG (500 mg) in distilled water (2 L) was placed in a beaker at the bottom of the chamber, and irradiated with DBD plasma for 20, 40, and 60 min, respectively. The aqueous solution, treated by the plasma, was evaporated to remove the solvent immediately.

### 3.3. Isolation of Oxidation Products

A sample solution of EGCG (500 mg) in distilled water (2 L) was irradiated for 20, 40, and 60 min, respectively, and the newly generated product was examined by HPLC. Among the dried reactant, the sample solution containing EGCG treated for 60 min showed the most improved effects toward *α*-glucosidase and *α*-amylase inhibition assays, with IC_50_ values of 20.1 ± 0.6 and 16.5 ± 0.7 μg/mL, respectively, compared with that of the parent EGCG. A solution containing EGCG, plasma-irradiated for 60 min, was suspended in 10% MeOH (200 mL), and the mixture was then partitioned with ethyl acetate (EtOAc) (200 mL × 3 times) to afford a dried EtOAc-soluble portion (384.8 mg). A part of the EtOAc-soluble portion (300.0 mg) was passed through a YMC gel ODS-AQ-HG column (1 cm i.d. × 41 cm, particle size 50 μm, YMC Co., Kyoto, Japan), and was then eluted with H_2_O in a stepwise gradient system containing increasing amounts of MeOH. The 15% MeOH solvent was eluted to yield pure compound **3** (*t*_R_ 8.0 min, 14.5 mg), the 30% MeOH solvent was eluted to yield pure compound **1** (*t*_R_ 15.0 min, 67.9 mg), and the 55% MeOH solvent was eluted to yield pure compound **2** (*t*_R_ 16.2 min, 1.5 mg). The chemical structures of these generated products **1**−**3** were determined by 1D-, 2D-NMR, and FABMS spectroscopy with a comparison to the reference data.

### 3.4. HPLC Analysis and Quantitation of Newly Generated Products

The reverse-phase HPLC instrument (LC-20AD, Shimadzu Co., Tokyo, Japan), equipped with a UV-photodiode array detector (UV-PDA, SPD-M20A, Shimadzu Co., Tokyo, Japan), was used for the chromatographic separation of plasma irradiation from EGCG. HPLC was performed on an ODS gel column (4.6 mm i.d. × 150 mm, particle size 5 μm, pore size 12 nm, YMC-Pack ODS-A, YMC Co., Kyoto, Japan). The mobile phase was composed of 0.1% HCOOH in H_2_O (solvent A) and MeCN (solvent B). A gradient solvent system was performed with a linear gradient from 5% to 100% on solvent B for 30 min. The flow rate and UV-PDA wavelength were 1.0 mL/min and 280 nm, respectively. The newly generated products from EGCG were monitored using their retention times (*t*_R_) and compared with original EGCG. Stock solutions of the four standards (EGCG, compounds **1**–**3**) were prepared in MeOH, each at 5000 mg/L. Working solutions were then obtained, as mixtures of these stock solutions after serial dilutions with methanol, to achieve five concentration levels in the rage of 500‒31.25 mg/L. The working solutions were filtered through a syringe filter (Fisher Scientific, Fair Lawn, MJ, USA) prior HPLC injection. After this, the linearity was determined by linear regression analysis of the integrated peak areas (*Y*) vs. the concentration of each standard (*X* mg/L) at five different concentrations (Appendix A Appendix A) [30].

### 3.5. Inhibitory Effects of α-Glucosidase and α-Amylase

The *α*-glucosidase inhibitory activities of the compounds were evaluated using a minor modification of a previously reported method [31]. The reaction mixture consisted of an enzyme solution (0.5 unit of *α*-glucosidase, 90 μL), a substrate (1 mM *p*-NPG, 100 μL) in a 0.1 M phosphate buffer (pH 7.0), and the isolated compounds, and positive control in 5% DMSO (10 μL). After incubation at 37 °C for 30 min, 0.1 M NaOH was added to quench the reaction. The amount of *p*-nitrophenol formed was measured at 405 nm using an ELISA reader (Infinite F200, Tecan Austria GmBH, Grödig, Austria). 

The α-amylase inhibition assay was performed using a previously reported method [32] with slight modifications. Porcine pancreatic α-amylase (1 unit, 90 µL) was incubated with the sample, and the positive control, at various concentrations. The reaction was started by adding 100 µL of a 1% potato starch solution, in 20 mM phosphate buffer (pH 6.9), to the reaction mixture. After incubation at 37 °C for 10 min, the reaction was stopped by adding a DNS color reagent solution (1% 3,5-dinitrosalicylic acid, 12% sodium potassium tartrate in 0.4 M NaOH). The mixture was then boiled at 80 °C for 10 min in a water bath and then cooled on ice. The α-amylase inhibitory activity was measured at 540 nm using an ELISA reader.
Enzyme inhibition (%) = [1−(*A*_1_/*A*_0_)] × 100(1)
where *A*_0_ and *A*_1_ are the absorbance of the control and sample, respectively. The α-glucosidase and α-amylase inhibition (%) was calculated, and the half-maximal inhibitory concentration (IC_50_) was determined by linear regression analysis of the inhibitory activities under the assay conditions. Acarbose was used as a positive control, and all assays were carried out in triplicate.

### 3.6. Statistical Analysis

Data for the in vitro analyses of *α*-glucosidase and α-amylase inhibitory activity were analyzed using the Proc GLM procedure of SAS software (version 9.3, SAS Institute Inc., Cary, NC, USA). The results are reported as the least square mean values and standard deviation. Statistical significance was considered at *p* < 0.05.

## 4. Conclusions

The present investigation evaluated the oxidation products of EGCG under aqueous medium, which enhanced the α-glucosidase and α-amylase inhibitory activities compared with the standard EGCG. The oxidized structures of the newly formed derivatives **1**–**3** were characterized using 1D-, 2D-NMR, and FABMS spectroscopic data and compared to the literature values. Among the oxidized products, EGCG quinone dimer A (**1**) exhibited the most potent inhibitory effects against α-glucosidase and α-amylase, with IC_50_ values of 15.9 ± 0.3 and 18.7 ± 0.3 μM, respectively. The most potent EGCG quinone dimer A (**1**) in the 60 min plasma-treated sample was 321.8 ± 1.2 mg/g, which agrees with the enhancement in terms of the anti-diabetic effects. A further systematic investigation with non-thermal plasma and its applications will be beneficial to the food industry in developing food safety and functionality.

## Figures and Tables

**Figure 1 molecules-26-05799-f001:**
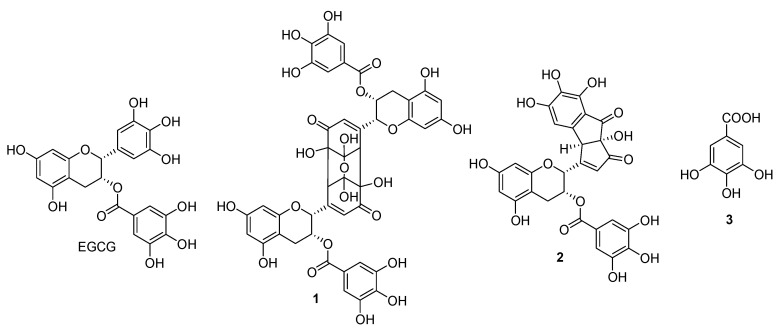
Chemical structures of oxidation products **1**−**3** of EGCG: **1**, EGCG quinone dimer A; **2**, theacitrinin A; **3**, gallic acid.

**Figure 2 molecules-26-05799-f002:**
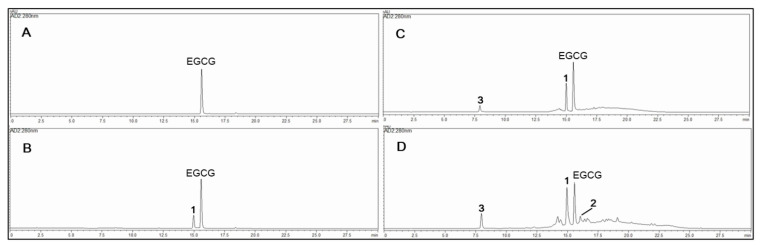
HPLC profiles of EGCG in aqueous solution by non-thermal plasma irradiation: (**A**) 0, (**B**) 20, (**C**) 40, and (**D**) 60 min. **1**, EGCG quinone dimer A; **2**, theacitrinin A; **3**, gallic acid.

**Figure 3 molecules-26-05799-f003:**
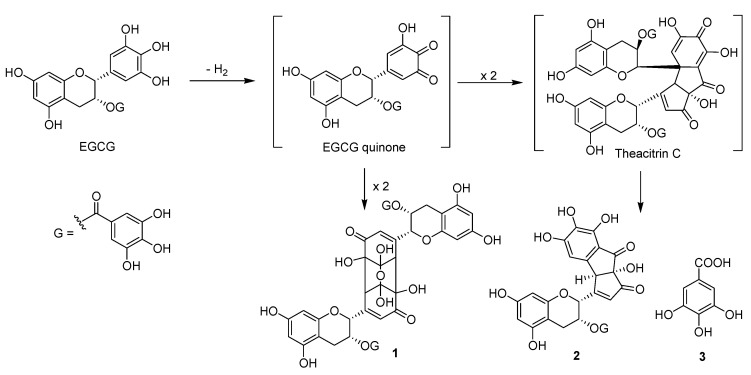
Proposed pathway for the oxidation of EGCG in aqueous solution by non-thermal plasma.

**Table 1 molecules-26-05799-t001:** Comparison of α-glucosidase and α-amylase inhibitory effects of plasma-irradiated EGCG for each time.

	IC_50_ Value (μg/mL) ^1^	
Plasma Irradiation Time (min)	α-Glucosidase	α-Amylase
0 (control)	44.5 ± 1.2 ^a^	34.5 ± 0.9 ^a^
20	40.5 ± 1.1 ^a^	29.5 ± 0.6 ^b^
40	28.6 ± 0.9 ^b^	19.2 ± 0.8 ^c^
60	20.1 ± 0.6 ^c^	16.5 ± 0.7 ^d^

^1^ All compounds were examined in triplicate. Different letters (a–d) within the same column indicate significant differences (*p* < 0.05).

**Table 2 molecules-26-05799-t002:** α-Glucosidase and α-amylase inhibitory effects of oxidation products **1**–**3**.

	IC_50_ Value (μM) ^1^	
Compounds	α-Glucosidase	α-Amylase
EGCG	97.2 ± 1.2 ^b^	75.5 ± 1.3 ^b^
**1**	15.9 ± 0.3 ^d^	18.7 ± 0.3 ^d^
**2**	20.7 ± 0.5 ^c^	25.3 ± 0.6 ^c^
**3**	>300 ^a^	>300 ^a^
Acarbose ^2^	169.0 ± 1.8 ^a^	70.7 ± 1.4 ^b^

^1^ All compounds were examined in triplicate experiments. Different letters (a–d) within the same column indicate significant differences (*p* < 0.05). ^2^ Acarbose was used as a positive control.

**Table 3 molecules-26-05799-t003:** Absolute content of individual components in the oxidized mixtures.

Compounds		Absolute Content (mg/g) ^1^		
	*t*_R_ (min)	20 Min-Reactant	40 Min-Reactant	60 Min-Reactant
EGCG	15.6	730.9 ± 2.1 ^a^	465.3 ± 1.5 ^b^	345.7 ± 1.1 ^c^
**1**	15.0	126.0 ± 0.9 ^e^	266.7 ± 1.1 ^d^	321.8 ± 1.2 ^c^
**2**	16.2	nd ^2^	nd	40.8 ± 0.2 ^f^
**3**	8.0	nd	65.7 ± 0.4 ^f^	101.5 ± 0.5 ^e^

^1^ All compounds were examined in triplicate experiments. Means with different letters (a–f) within the column differ significantly (*p* < 0.05). ^2^ nd: not detected.

## Data Availability

The data presented in this study are available on request from the corresponding author.

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
