# Peer review of "Plasma-Induced Oxidation Products of (–)-Epigallocatechin Gallate with Digestive Enzymes Inhibitory Effects"

_molecules, 2021, doi:10.3390/molecules26195799_

Round 1

Reviewer 1 Report

This manuscript reported the generation of the plasma-induced oxidation products of EGCG and their enzymes inhibitory activities. Initially, three degradation products of EGCG were generated by incubation the EGCE with plasma, their identity and the amount was analyzed by the LC, NMR, etc. Later, both of the reaction mixture and isolated products were submitted for the enzyme inhibitory activities, and it was concluded that compound 1, which is a dimer displayed the best activity among the all. Overall, this manuscript is a natural product paper, focusing on the activity of the derivative of the natural product. As far as I am concerned, it can be assigned as major revision.

Here are the questions and concerns.

  1. For the experiment design, why the treatment time stopped at 60 min? It seems that at 60 min, the new metabolites just started forming, especially compound 2 is just only one of those metabolites.

  1. At the 40 min time point, free Gallic acid was observed, what and where is the rest of the part of EGCG? It should display in the chromatogram.

  1. Temperature issue? It is not clear that which temperature the reaction was performed. Will the temperature affect the formation of the EGCG derivatives?

Author Response

SEPTEMBER 15, 2020

Dear Editor,

Molecules

Attached please find a revised manuscript entitled “Plasma-induced oxidation of (–)-epigallocatechin gallate with digestive enzymes inhibitory effects” by Tae Hoon Kim et al.

Thank you for your kind and valuable comments to our manuscript. According to the valuable suggestions and comments, we have revised our manuscript and attached detail. Changed words were expressed red ones. Please consider our following responses to each suggestion and comment.

Sincerely,

Tae Hoon Kim,

+82-53-850-6533 (Ph)

+82-53-850-6539 (Fax)

Reviewer 2 Report

Comments to the Author

This study isolated and characterized three known oxidative products of EGCG under the condition of DBD plasma. Authors further evaluated the inhibitory effects of these oxidative products against α-glucosidase and α-amylase compared to EGCG, as a result that product 1 had the most potency. However, one major issue of this study is the lack of novelty and it seems relatively superficial. On one hand, the inhibitory effects on enzymes were evaluated only by IC50 values in vitro, which needs to be systematically clarified by molecular docking, cellular model or animal models by multiple technical methods. On the other hand, few supports were provided in the manuscript to demonstrate the molecular mechanisms for this experimental phenomenon.

Other main comments for the authors’ attention are the following:

1 Tab.1: What is the concrete meaning of “a-c” of “w-z”? The control group was also marked with significant differences. Why?

2 The resolution of Fig.2 needs to be improved.

3 The reliability of the quantitative methods was not mentioned in the manuscript.

4 Tab.3: The unit of quantitative results should be reflected in the table.

5 Fig.3: The enzymes involved in the production of these oxidative products should be indicated.

Author Response

(The authors gave the same response as above.)

Reviewer 3 Report

Please refer to the observations/suggestions available as notes in the enclosed file.

Author Response

(The authors gave the same response as above.)

Round 2

Reviewer 1 Report

The author addressed my questions and concerns, thus I agree to accept it in the present form.

Reviewer 3 Report

The authors addressed all the previous observations.

There are still some typos (i.e. check the chemical formulas, stoichiometric coefficients should be subscript) which could also be modified during the proofreading at the discretion of the editor